# The Use of Lasers in Dental Materials: A Review

**DOI:** 10.3390/ma14123370

**Published:** 2021-06-18

**Authors:** Emmanouil-George C. Tzanakakis, Evangelos Skoulas, Eudoxie Pepelassi, Petros Koidis, Ioannis G. Tzoutzas

**Affiliations:** 1Department of Operative Dentistry, School of Dentistry, National and Kapodistrian University of Athens, 11527 Athens, Greece; tzoudent@dent.uoa.gr; 2Institute of Electronic Structure and Laser, Foundation for Research and Technology-Hellas, 70013 Heraklion, Greece; skoulasv@iesl.forth.gr; 3Department of Periodontology, School of Dentistry, National and Kapodistrian University of Athens, 11527 Athens, Greece; epepela@dent.uoa.gr; 4Department of Prosthodontics, Aristotle University of Thessaloniki, 54124 Thessaloniki, Greece; pkoidis@dent.auth.gr

**Keywords:** laser, zirconia, surface texturing, sintering, titanium, surface micro-topography

## Abstract

Lasers have been well integrated in clinical dentistry for the last two decades, providing clinical alternatives in the management of both soft and hard tissues with an expanding use in the field of dental materials. One of their main advantages is that they can deliver very low to very high concentrated power at an exact point on any substrate by all possible means. The aim of this review is to thoroughly analyze the use of lasers in the processing of dental materials and to enlighten the new trends in laser technology focused on dental material management. New approaches for the elaboration of dental materials that require high energy levels and delicate processing, such as metals, ceramics, and resins are provided, while time consuming laboratory procedures, such as cutting restorative materials, welding, and sintering are facilitated. In addition, surface characteristics of titanium alloys and high strength ceramics can be altered. Finally, the potential of lasers to increase the adhesion of zirconia ceramics to different substrates has been tested for all laser devices, including a new ultrafast generation of lasers.

## 1. Introduction

The application of lasers is wide, ranging from basic science and industry, to medicine and manufacturing industry. Nowadays, typical laser source scan deliver extreme amounts of energy in such confined spaces, restrained only by the diffraction limit of converging optics and laser frequency. The successful energy deposition, direct on controllable space as well as the independence of the material case, has made lasers a unique and versatile tool for variable applications such as cutting, welding, soldering and surface functionalization [1]. These advancements led the laser utilization on medical materials for clinical applications and research where, so far, remarkable advancements for most medical specialties, and of course dentistry, have been made [2,3].

The use of lasers in dentistry is not a revolutionary method since they have been introduced in almost all dental specialties for more than two decades [2,3], working as a diagnostic tool for the detection of caries, of subgingival calculus, as a cutting tool for hard dental tissues [2] and as a disinfecting tool of root canals [3]. Moreover, the anxiety and dental fear of many patients towards dental rotary cutting devices without employing injectable local anesthesia was minimized when hard dental tissue laser applications were introduced [4].

Nowadays, laser technology is used in clinical applications [5,6], mainly for the management of hard and soft tissues, and in the field of dental materials [7]. Lasers offer new approaches to the elaboration of dental materials, such as metals, ceramics and resins that require high energy levels and delicate processing. Many traditional methods were displaced and the management of hard and sensitive materials was facilitated [6,7]. Moreover, the use of ultrafast lasers in dentistry seems promising since they allow surface processing even for very hard ceramics, such as zirconia, with minor structural changes [8]. The aim of the present review study is to thoroughly analyze the use of lasers in the processing of dental materials. Laser assisted engineering of dental materials includes cutting, welding, sintering and texturing. These applications are thoroughly discussed combined with contemporary dental materials. New generations of intraoral laser devices might be a future project.

## 2. Energy Absorption and Mechanism in Materials and Tissues

The direct irradiation of a solid surface with a laser source normally leads to energy absorption. The amount of the absorbing energy as well as the absorption mechanism is highly dependent on the characteristics and the nature of the material, along with the laser specifications (pulse duration, wavelength, repetition rate, etc.). Furthermore, the development of intense ultrashort laser pulses with duration in the femtosecond regime has made a huge contribution to dental material processing. This is due to the fact that the energy deposition can take place before the carrier system relaxation [9]. In principle, this means that the energy is deposited in much shorter timescales than the electrons/carriers need to render the absorbed energy in the material lattice which will eventually lead to a rapid heat increment.

Depending on the irradiation parameters, direct material removal may occur [10] along with phase transition (melting) [11,12], or even controllable surface modification [13,14]. As far as metals are concerned, the energy absorption is dominated by free carrier absorption, as the electrons in the conduction band directly absorb photons and transit in higher energy states. On the other hand, for wide-band gap materials, such as ceramics or polymers, absorption of laser light can be possible only when high intensities are applied, giving rise to nonlinear optical processes and multiphoton absorption [15,16].

Concerning tissue ablation, taking into consideration the chemical complexity of each tissue type it is extremely difficult to provide a model for the ablation of complex organic molecules. However, there are studies that consider that ablation occurs by the heating and evaporation of water concentration in the tissue [17] or even the tissue composition and morphology [18].

There is no doubt that the excitation dynamics are rather complex and hard to analyze for each dental material separately. However, several standard timescales of phenomena and processes taking place during and after the irradiation of a solid material with an ultrashort laser pulse have been described and along with a graphical illustration of the temporal distribution of a single laser pulse with fs, ps and ns pulse widths are presented at Figure 1. From the moment a laser pulse strikes the surface, electron/carrier excitation takes place almost instantly in the femtosecond regime. The timescale of material melting ranges among every material case and it can be roughly placed in the picosecond regime. Finally, ablation and other morphological changes can last up to the nanosecond regime while.

## 3. The Use of Lasers in Clinical Dentistry

Lasers are used in most areas of clinical dentistry and their use is steadily increasing. Their application includes caries detection, caries removal, dental cavity preparation, dentine hypersensitivity management, tooth bleaching, photodynamic therapy, excisional biopsy, aphthous ulcers management, frenectomy, vestibuloplasty, removal of irritation fibroma, removal of hyperplastic tissues, haemangioma, exposure of impacted teeth, gingivectomy and gingivoplasty, apicoectomy, an adjunct to endodontic treatment [19], gingival melanin hyperpigmentation removal [20,21,22], non-surgical periodontal treatment of untreated periodontitis [23,24,25,26,27,28,29,30] and treatment of peri-implant diseases [31,32,33]. In terms of endodontic treatment, lasers are used for post-chemomechanical preparation to reduce the remaining bacterial load either by direct irradiation of the canal walls or by laser assisted irrigation via erbium laser group. In terms of gingival melanin hyperpigmentation removal, a recent systematic review identified the use of two laser wavelength groups-near infrared diode and erbium group of mid-infrared lasers and failed to draw conclusions on the optimal laser group [22].

In untreated periodontitis patients, lasers have been studied both as monotherapy and as an adjunct to conventional non-surgical periodontal treatment (or non-surgical mechanical instrumentation). In terms of non-surgical periodontal treatment, laser use aims mainly in root surface detoxification. A recent systematic review and meta-analysis showed that in untreated periodontitis patients, laser monotherapy leads to similar clinical improvement to conventional non-surgical periodontal treatment alone [29], which is in agreement with earlier reviews [27,34]. Systematic reviews on untreated periodontitis failed to show further clinical benefits with the addition of laser treatment to conventional non-surgical periodontal treatment [27,28,30]. Specifically, a 2015 systematic review and meta-analysis found that, when used adjunctively to conventional non-surgical periodontal treatment, neither the diode nor the Nd:YAG laser achieves additional clinical benefits beyond that achieved by conventional non-surgical periodontal treatment alone [28]. Then, a recent systematic review showed that the adjunctive use of lasers to conventional non-surgical periodontal treatment does not lead to superior clinical improvement as compared to conventional non-surgical periodontal treatment alone [30]. Nowadays, lasers are used as an adjunct to conventional non-surgical periodontal treatment in periodontitis. It should be stressed that the routine use of any laser for the treatment of periodontitis cannot be suggested [25]. Decontaminating (or detoxifying) the oral biofilm-contaminated titanium surface without altering it is fundamental for the treatment of periimplantitis [35]. Laser treatment has been proved to be effective in the decontamination of oral biofilm-contaminated titanium surfaces [36,37]. Several types of lasers have been studied for this purpose. Among them, Er:YAG seems to be the best for titanium surface decontamination without damaging the titanium surface [26,36,38]. Specifically, Er:YAG laser and Er, Cr:YSGG laser treatment have been found to be effective methods for the detoxification of oral biofilm-contaminated titanium surfaces without surface alterations [36]. In terms of clinical benefit, a systematic review and meta-analysis showed that laser treatment resulted in similar clinical improvement as compared to conventional implant surface decontamination methods [31]. Another systematic review and meta-analysis showed that the use of lasers is not superior to conventional therapeutic approaches for the treatment of peri-implantitis [32]. An American Academy of Periodontology systematic review showed that lasers as an adjunct to surgical/non-surgical treatment of peri-implant mucositis and peri-implantitis offer minimal clinical benefit [33]. It seems that laser treatment is promising for the management of peri-implantitis, though further research is required to draw safe conclusions.

## 4. The Use of Lasers in Dental Materials

Apart from the application of lasers in clinical dentistry, lasers are used in the processing of dental materials as well (Table 1). In terms of dental materials processing, lasers are mainly used for cutting, welding, melting and sintering as well as for texturing [39].

### 4.1. Lasers for Cutting

The use of lasers in the field of cutting and welding is increasing displacing older traditional methods [4]. From an industrial point of view, several laser cutting techniques have been suggested. Cutting metals based on the principle of melting is related to internal gas production. The laser causes melting while the gas removes the molten material and thus material sectioning is initiated. For polymeric materials, the gas used is compressed air. Other cutting methods are based on evaporation or chemical degradation [40].

The advantages of laser cutting over mechanical cutting lie on the reduced contamination of the workpiece, since there is no cutting edge which can be contaminated by the material or contaminate the material and on the fact that the laser beam does not wear out during cutting (as it happens with cutters) [39]. Moreover, there are reduced chances of warping the material being cut, as laser systems have a small range of heat affected bands. The main disadvantage of laser cutting is the high energy consumption. The power consumption and efficiency of a specific laser varies depending on the output power and operating parameters. The amount of power that laser cutting requires is known as heat input, which depends on the type of material, thickness, process (reactive/inert) used, and the desired cutting rate. In clinical dentistry, lasers, based on their cutting ability, are used for removing hard dental tissues and dissecting soft tissues. Lasers can be used for cutting ceramic materials or alloys, either as the only cutting method or in combination with other methods [4,41]. In terms of zirconia, it is very difficult to cut it at low temperatures. Its processing capacity is significantly improved at increased temperature. Heating zirconia by using a laser beam significantly reduces the cutting difficulty. The combined use of laser and conventional cutting is beneficial since it reduces the wear of cutting tools and allows to use cutting instruments of cost lower than that of diamonds, such as carbides and nitrides [41]. Especially for zirconia ceramics, powerful lasers have been developed which allow the creation of complex shapes, frames and other structures by cutting a flat surface material (bulk material). This laser belongs to the category of femtosecond lasers (ultrafast lasers). Thus, a crown is manufactured in approximately an hour by using a 1 cm^3^ zirconia cube [42]. Laser straight cutting of zirconia by using CO_2_ laser resulted in cut section free of major cracks [43]. With laser-assisted grinding (Yb-fiber laser 1070 nm) of fully sintered zirconia, both the grinding force and the tool damage were reduced. However, microcracks on the zirconia surface were observed implying that additional polishing or fine grinding is required to reduce subsurface cracks [44].

In terms of cutting and removing composite resin from a tooth, Er:YAG lasers might cause photoreceptor-type damage. Thus, lasers successfully remove the superficial layers of the resin causing melting in its mass and sublimation (Figure 2), but they hardly remove the resin from the bottom of the cavity without affecting the underlying dentin (Figure 3). Er:YAG laser-assisted resin removal is an alternative method to resin removal by using rotary cutting instruments, though it cannot displace the conventional cutting means [45]. In terms of the time required to remove the resin from a resin-filled dental cavity, it was found that the complete elimination of the material from the walls of the cavity was more time consuming for the laser-ablated groups. For this purpose, Er:YAG laser was efficient without raising the temperature and causing thermal damage to the dental pulp [46]. In terms of removing fiber posts cemented with composite resins, the Er:YAG laser, compared to ultrasonics, was faster and led to less temperature increase at the root surface [46]. It seems that Er:YAG laser might be an alternative to sonication for post removal [47].

Another recent application of lasers in orthodontics is the softening and complete composite resin removal from de-bonded orthodontic brackets [48]. It seems that thermal degradation of composite material is a safe procedure that can substitute traditional rotary instruments, without damaging the underlying sound enamel structure or increasing intrapulpal temperature [49,50].

### 4.2. Lasers for Welding

Cutting the metal framework after casting, due to unsatisfactory fit in implant or dental abutments, is common in clinical dentistry. The main cause is the distortion of the wax pattern or deformation or stresses after casting [39]. Welding the dissected parts of the metal framework has been used to solve problems related to laboratory distortion which results in metal framework misfit in the marginal area. Welding the metal parts reduces the distortion of the framework and improves the fit to the abutments and the uniform stress distribution [39].

Various methods have been used for connecting the parts of the metal framework. Among them, the laser-assisted welding is preferred, since it allows the alloy to melt and integrate into the original framework by creating a new joint [39,51]. Specifically, when the light beam reaches the surface of the metal, the metal absorbs it, converting it into heat which then penetrates the metal, due to conductivity. The metal melts, as a result of the high heat concentration, and forms a hollow cavity that will then be filled with molten metal [52]. The industrial improvement of small powerful units of pulsed Nd:YAG lasers has made this technological achievement even more efficient. Laser-assisted welding is precise and well outlined due to the concentration of energy in a very small part of the framework. Furthermore, the operator controls each stage of the welding process at any time. Any distortion that might occur during the welding process can be immediately detected and corrected by adding new spot welds on the opposite side of the initial distortion point [53]. For the dental lab, it seems that laser welding provides precision, reduced distortion and a minimal heat affected zone [54].

Concerning intraoral welding by using Nd-YAG laser, in a case report a metallic (Co-Cr-Mo) bar for a maxillary overdenture was welded intraorally in 47 s without patient reported pain or discomfort [55]. In Orthodontics, Nd-YAG laser is a useful alternative solution for welding orthodontic wires [56].

### 4.3. Lasers for Melting and Sintering

Laser sintering is used for manufacturing metal frames of metal-ceramic restorations. It aims at the elimination of the negative consequences of casting and at the reduction of financial cost and labor time. The porosity of the material, the fit in the marginal area and the mechanical properties in general are of outmost importance for the success of the reconstruction. Selective laser sintering is a layered manufacturing process that allows the creation of complex three-dimensional parts by solidifying successive layers of material powder on top of each other. It is achieved by treating the selected areas using the thermal energy provided by a focused laser beam. Using a beam deflection system (galvanic mirrors), each layer is scanned based on the respective cross section, as assessed by the CAD model. The deposition of successive layers of material (20 to 150 μm in thickness) is carried out by using a powder deposition system [57,58].

The advantages of laser sintering include the rapid construction of complex metal frameworks and the operation of an automatic system as well as the reduced working time, since several stages are skipped, such as wax model making, coating, dewaxing and casting. It saves on the use of the alloy by reducing metal waste, since the required amount of the alloy is precisely selected and controlled. Laser sintering requires expensive equipment, which is a disadvantage of the technique. Proper fit of the restoration at the marginal area is fundamental for the success of the dental restoration. An in vitro study compared marginal fit and axial wall adaptability of cobalt-chromium copings that were fabricated by using either the conventional lost-wax or the metal laser sintering technique and found significantly higher marginal fit and axial wall adaptability for the laser sintering than for the lost-wax technique, though for both techniques the values of marginal fit were within the clinically acceptable limit (<50 μm) [58]. Another similar in vitro study, where the marginal and internal fit of cobalt-chromium copings that were fabricated by using the conventional lost-wax or the direct metal laser sintering technique, showed similar marginal fit for both methods but higher internal fit for the lost-wax than the direct metal laser sintering technique [59].

### 4.4. Lasers for Texturing

Lasers are a successful method for modifying the morphology of the surface of biomaterials. They are advantageous due to high resolution, high operating speed and low operating cost. Laser-assisted surface modifications usually range from coating with bioactive materials (such as calcium phosphate or bio glass) [60,61] to the production of highly controlled macro- and micro-structures (via the laser texture technique) [62].

In terms of titanium, which has several applications in dentistry, laser treatment of pure titanium surface seems to improve its mechanical properties. The mechanism by which this improvement is achieved is related to the entry of residual stresses due to the rapid melting-coagulation by the application of the laser [63]. Titanium alloys are the main dental implant material. The nanotopography (with a scale ranging from 1 to 100 nm) of dental implants seems to affect cell-implant interactions at the cellular and protein level [64]. Implant surface energy increase depends not only on surface roughness but on surface chemistry as well [65]. Implant surface nanotopography changes affect the surface physically, chemically and biologically [66]. Specifically, it has been found that the adhesion of osteoblasts was increased on nanophase metals (metals that possess particle or grain sizes less than 100 nm) compared to conventional metals, including Ti and Ti6Al4V [67]. This might positively affect osseointegration. Based on these findings, an implant has been created with collar surface treated by laser micromachining to generate nano-channels (Laser-Lok^®^, BioHorizons, Birmingham, AL, USA) [68,69]. Thus, Laser-Lok^®^ is a series of precision-engineered cell-sized channels laser-machined onto the surface of dental implants and abutments. Connective tissue attachment formation has been found histologically around Laser-Lok abutments [70]. For Laser-Lok, the connective tissue fiber direction in the soft tissue attachment was perpendicular to the titanium implant surface, which differs from the parallel fiber orientation usually seen [71,72]. Moreover, bone-to-implant contact was significantly improved for Laser-Lok compared to a turned titanium implant surface [70]. Beneficial effects on crestal bone preservation have been found for Laser-Lok [73]. It seems that findings on laser-modified implant surfaces are encouraging, though further research is required for safe conclusions.

## 5. Zirconia and Lasers

High strength ceramics have been recently introduced in restorative dentistry with many drawbacks. Several techniques have been suggested in order to increase cement attachment to these materials [74]. Laser applications (Table 2) are alternative methods to improve the bonding of reinforced ceramics with resin cements [75]. A significant factor is that high strength ceramics do not effectively absorb certain wavelengths, such as the Nd:YAG laser (λ = 1064 nm) do. In order to increase energy absorption, graphite powder is applied to the ceramic surface. During laser application, the toner is removed from the surface with small explosions [76].

Based on findings where the action of the Nd:YAG laser had a similar effect to hydrofluoric acid (HF),by increasing the surface irregularities and adhesion of composite resins, researchers [76,77] found a positive effect of Nd:YAG after sandblasting (50 μm) and silanization to bond strength in In-Ceram zirconia core material. A scanning electron microscopy (SEM) study showed the removal of material from the surface due to the micro-explosions caused by the laser and consequently the creation of gaps. Moreover, there was diffusion and melting of the superficial layer creating a smooth cell-like surface [76]. When the local temperature exceeds the melting point of zirconia (2700 °C), the surface expands and then immediately shrinks during solidification. The stresses generated by the temperature change explain the cracks of the ceramic surface [78]. Nd:YAG settings have been based on previous research in an alumina ceramic [77]. The most effective settings seem to be 2 W (100 mJ/pulse, 20 Hz, 2 min), where surface bubble structures and microcracks are created without affecting bond strength [79]. However, Yttria-stabilized zirconia materials are considered a suitable substrate for this laser due to their high hardness and low coefficient of thermal conductivity [80]. It was found that Nd:YAG laser improved both the roughness and the strength of the bond, though the point of application left a silvery tinge [76]. Similar results were found by others for shear bond strength [81,82]. The surface cracks, the darkening of the surface and the reduction of the oxygen concentration on the surface of the zirconia as well as the reduction of the mechanical strength of the material observed during the use of Nd:YAG for welding zirconia and zirconia toughened alumina frameworks make it unsuitable for this procedure [83]. When laser and sandblasting were combined, Nd:YAG improved the bond and sandblasting increased the roughness [79]. However, another study reported higher roughness with laser alone (without toner application) [78].

The CO_2_ laser is suitable for ceramics because its emission wavelength (2.3–10.6 μm) is absorbed by the ceramic. When used on a ceramic surface, concave “tears” are formed, which is typical of surface heating. They seem to increase micromechanical retention. Improved laser bond has been reported experimentally due to microcracks created by areas of restraint capacity [84]. Surface roughness did not seem to be significantly affected by CO_2_. Though, with laser application at 4 W setting for 50 s, the surface was rough with gaps and had the appearance of scaly plates and the bond with resin cements was improved [79]. At 4.5 W, for 60 s the roughness and the groove depth were increased [85]. At 5 W, significantly more and wider cracks were created by Nd:YAG. When the shear strength was evaluated with settings ranging from 2 to 5 W, it was increased at 2 W and decreased at 5 W [86]. The CO_2_ laser releases more energy than Nd:YAG due to increased beam thickness (1 mm versus 320 μm), and thus it cannot develop a thin retentive structure on the surface of zirconia [78]. In another study, 2 W for 10 s of CO_2_ laser increased shear bond strength (SBS) to resin cements [87]. Similar results with different settings 20 W/10 mJ have been reported [88].

An Er:YAG laser has a wavelength that coincides with the maximum absorption by water and is well absorbed by the hydroxyls of hydroxyapatite [89]. It removes granules by micro-explosions and by evaporation, a process called ablation. The effect of the Er:YAG laser on the roughness of two Y-TZP ceramics was studied in relation to sandblasting. Covering the zirconia surface with graphite is considered necessary for energy absorption, although hydroxyapatite powder has been suggested as an alternative to graphite. At high settings (at 600 mJ), there was extensive material destruction and increased roughness, while at low settings (200 or 400 mJ) the results were similar to those of sandblasting [89]. Moreover, the laser caused color changes on the zirconia surfaces, as assessed by optical microscopy. With SEM, smooth surfaces were observed surrounded by surface cracks of increasing intensity depending on the laser density. At high settings (400, 600 mJ), melting of the superficial zirconia layers, significant loss of mass and deep cracks were observed [89]. At different settings (150 mJ, 1 W, LP for 20 s), Er:YAG improved the bond and the resistance to microleakage [81,90]. At 2 W, the roughness was similar to sandblasting and the bond strength was lower to sandblasting group, while at 3 and 4 W large cracks and material loss occurred [79]. Irradiation time seems to play an important role, since at 5 and 10 s the results differed [79,86]. Another study found lower roughness with Er:YAG laser application at 400 mJ as compared to 110 μm Al_2_O_3_ sandblasting or 30 μm silica coating [91]. It has been suggested that the application of Er:YAG at 400 mJ might be an alternative to sandblasting with 110 μm [92]. It should be stressed that Er:YAG laser has a small depth of effect on zirconia (certain μm), thus structural changes are limited to the outer surface of the material [93]. It was reported that Er:YAG laser of 6 W for 5 s increased shear bond strength (SBS) [87]. Moreover, Er,Cr:YSGG laser at 3 W seems to increase μSBS of resin cement to zirconia surface [94]. However, negative results were found in studies reporting that Er:YAG laser irradiation does not improve the bonding property of zirconia ceramics to resin cement [95,96]. Overall intraoral laser devices cannot guarantee satisfactory long term adhesive bond to resin materials without material deterioration [74,97]. High-speed pulse lasers (femtosecond lasers, Ti: sapphire) have been tested for zirconia implant surface treatment and zirconia frameworks with promising results [42,98,99], which seem to change both the roughness and the texture of the zirconia surface [100], increasing adhesive potential to zirconia [99,100]. It was found that by decreasing the angle between the ceramic surface and the laser beam, the bond strength increased, and that the optimum angle was 45° [99]. Femto laser patterning of the zirconia surface is possible, initiating future trends to optimize bonding to different substrates, providing rough repetitive retention grooves with less phase transformation than aggressive sandblasting pretreatment [8].

Moreover, in high-strength materials, like zirconia, multiple designs and surface patterns can be beneficial for the retention and adhesion of dissimilar materials. Parallel and repetitive grooves increase the retention of both resin composite and ceramic materials [8]. Although shear bond strength (SBS) did not yield statistically significant differences [101], future research might optimize compatibility and might explore adhesive potentials. More retentive patterns (Figure 4) or designs with less destructive concept (Figure 5) or even more complicated designs with subsurface undercuts might optimize bond strengths.

## 6. Conclusions

Based on the present review study, the following conclusions can be drawn:Intraoral lasers cover a wide range of applications in clinical dentistry.Cutting of metals, polymers and zirconia ceramics is facilitated by specific laser devices.Metal processing benefits from accurate laser welding procedures.Increased accuracy and reduced working time are the main advantages when manufacturing base metal frameworks by laser sintering procedures.Optimized retentive patterning of titanium and the zirconia surface by laser patterning is thoroughly investigated.

## 7. Future Perspectives

The role of lasers in clinical dentistry is well established. Lasers are applied in almost all areas of clinical practice, ranging from diagnosis to hard tissue cutting and surgical intervention, and in the processing of dental materials. The property to deliver high concentrated power eliminates the negative consequences of traditional techniques in processing hard materials. Dental technicians have almost substituted traditional soldering procedures with laser-assisted welding. Construction of metal frameworks following laser sintering might eliminate waxing and casting in the near future. Laser modification of surface characteristics by altering roughness and creation of retentive patterns is also a promising research trend for both titanium alloys and high strength ceramics.

The numerous laser devices available for intraoral and laboratory use differ in emitted wavelength, power, intensity, fluence and pulse duration. Although a wide range of devices is available for dental applications, some industrial laser devices seem to be more effective in very hard material processing, such as with zirconia. The ability of ultrafast lasers to ablate and cut with outstanding precision the yttria stabilized zirconia (YTZP) surface, creating a wide range of spots or lines, warrants a research strategy in the future. Such laser devices are expensive, which is a serious limitation reducing their use in the dental lab. Moreover, the fabrication of intraoral ultrafast laser hand pieces (of minimal dimensions) with reasonable cost seems to be a future project.

## Figures and Tables

**Figure 1 materials-14-03370-f001:**
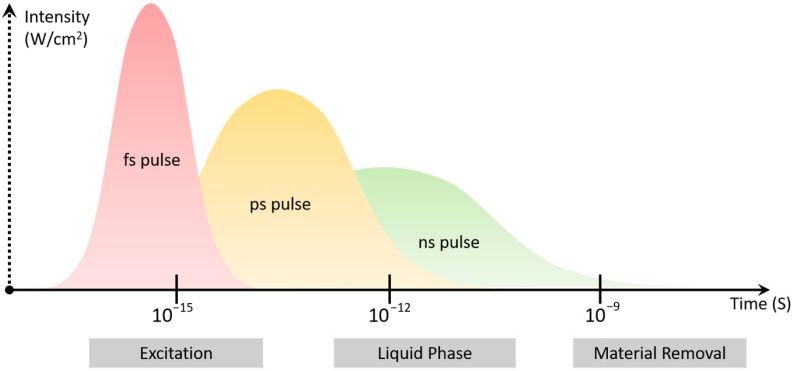
Simple schematic illustrating a femtosecond, picosecond and nanosecond laser pulse distributed in time and intensity along with the basic physical processes that occur during laser induced material removal.

**Figure 2 materials-14-03370-f002:**
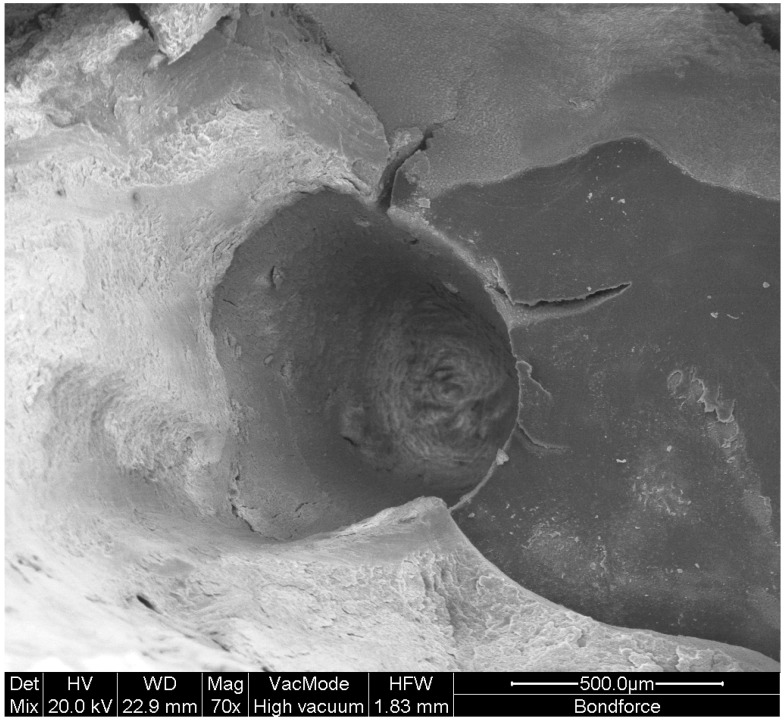
Composite resin crater created during Er-YAG laser application for resin restoration removal (concession by I. Tzoutzas).

**Figure 3 materials-14-03370-f003:**
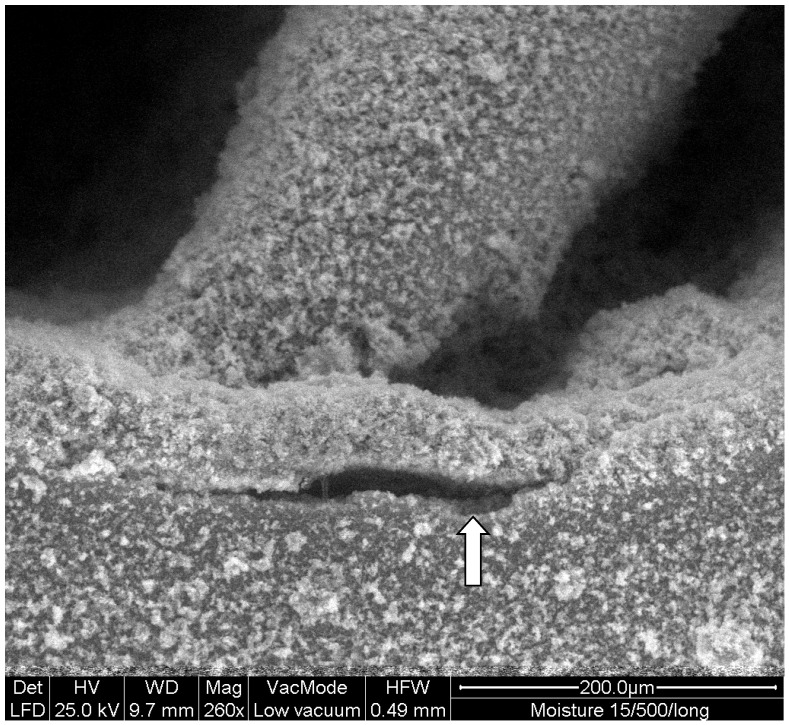
Composite resin marginal detachment from dentin caused by Er:YAG laser irradiation after incomplete resin restoration removal. White arrow indicates the damaged dentin surface (concession by I. Tzoutzas).

**Figure 4 materials-14-03370-f004:**
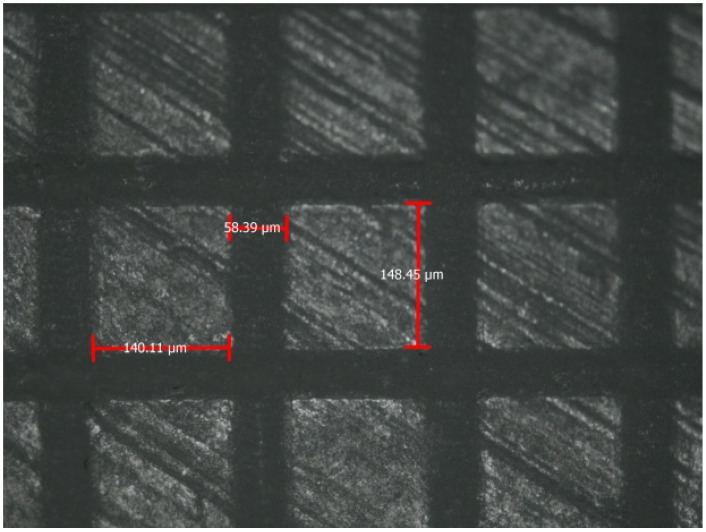
Horizontal and vertical grooves (50 μm depth, 50 μm width) after femtosecond laser application in a flat Y-TZP zirconia ceramic surface [8].

**Figure 5 materials-14-03370-f005:**
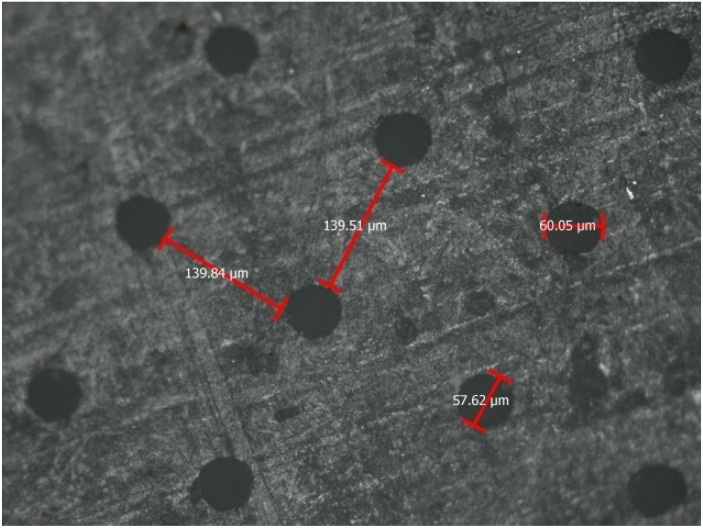
A different pattern constructed by femtosecond laser application in zirconia surface. Dark bullets caused by laser drilling creating minimal circular retentive areas to enhance micro-retention [8].

**Table 1 materials-14-03370-t001:** Laser applications in contemporary dental materials.

Dental Material	Laser-Assisted Engineering			
	Cutting	Welding	Sintering	Texturing
Zirconia Ceramics	+	−	−	+
Composite Resins	+	−	−	−
Titanium Alloys	−	−	−	+
Base Metal Alloys	−	+	+	−

**Table 2 materials-14-03370-t002:** Contemporary laser dental materials processing.

Laser Devices	Dental Material		
	Zirconia Ceramics	Composite Resins	Base Metal Alloys
CO_2_	+	−	−
Er:YAG/Er,Cr:YSSG	+	+	−
Nd:YAG	+	−	+
Ti:sapphire (femto)	+	−	−
Yb:KGW (femto)	+	−	−

## Data Availability

Data sharing not applicable. No new data were created or analyzed in this study. Data sharing is not applicable to this article.

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
