# Peer review of "The Use of Lasers in Dental Materials: A Review"

_materials, 2021, doi:10.3390/ma14123370_

Round 1
Reviewer 1 Report
The authors provide an overview of the use of lasers in clinical dentistry and dental materials in this review paper. The manuscript focuses on the specific area of interest, and it offers the recent trend and research direction of the field. However, for the publication in Materials, it is recommended to supplement the manuscript by providing more detailed contents as listed below.
[1] In section 3, please provide more details on the mechanisms, methodologies (e.g., wavelength of light and duration), and generally reported pathological effects for different dental diseases.
[2] In section 4, please discuss more examples of laser-assisted engineering of dental materials with graphical illustrations to help the understanding of potential readers. It would be good to cite figures in previously reported papers.
[3] The text mentions Figure 4a,b and Figure 5a,b, but there is only one image in Figure 4 and 5.
[4] Section 3 focuses on the skeptical sides of the use of lasers in clinical dentistry, but it is not consistent with what is described in Conclusions and Future perspectives. Please provide a clear view of this part.
Author Response
Reviewer 1
[1] In section 3, please provide more details on the mechanisms, methodologies (e.g., wavelength of light and duration), and generally reported pathological effects for different dental diseases.
In section 3, changes have been made based the reviewer’s suggestions. Further details on the use of lasers in clinical Dentistry (e.g. wavelength of light and duration, generally reported pathological effects for different dental diseases) are beyond the scopus of the present article.
[2] In section 4, please discuss more examples of laser-assisted engineering of dental materials with graphical illustrations to help the understanding of potential readers. It would be good to cite figures in previously reported papers.
We thank the reviewer for the suggestion, in order to help the potential reader we inserted two Tables matching dental materials, laser devices and their applications in Dentistry to summarize all the information provided in section 4 and 5. The figures 2-5 are the most representative for resin and zirconia materials laser processing and are part of original unpublished work of professor Dr. Tzoutzas I. and Dr. Tzanakakis E.
[3] The text mentions Figure 4a,b and Figure 5a,b, but there is only one image in Figure 4 and 5.(corrected)
[4] Section 3 focuses on the skeptical sides of the use of lasers in clinical dentistry, but it is not consistent with what is described in Conclusions and Future perspectives. Please provide a clear view of this part.
In section 3, the clinical use of lasers is analysed. Lasers are used in most areas of clinical Dentistry and their use is steadily increasing. They are mostly used as an adjunct to conventional dental and periodontal treatment rather. Their clinical use as monotherapy is limited to few therapeutic procedures. Nowadays, their use in clinical Dentistry can not rule out conventional therapeutic techniques. Moreover, when used as adjuncts to conventional treatment, their additive benefit is questioned for certain therapeutic techniques. Based on these findings, section 3 presents (a) their clinical use and (b) the “skeptical sides” of the use of lasers in clinical dentistry concerning their use as monotherapy and their additive effectiveness when used as adjuncts. It should be taken into consideration that the aim of the present review study is to thoroughly analyze the use of lasers in the processing of dental materials. The “positive” conclusions and the promising future perspectives concern the use of lasers in the dental materials processing, which is the aim of the present article. Therefore, there is no inconsistency between section 3 and conclusions/future perspectives.
Reviewer 2 Report
The presented publication takes up a current hypothesis the Lasers have been well integrated in clinical Dentistry for the last two decades providing clinical alternatives in the management of both soft and hard tissues with an expanding use in the field of dental materials. One of their main advantages is that they can deliver very low to very high concentrated power at an exact point on any substrate by all possible means. The aim of this review was to thoroughly analyze the use of lasers in the processing of dental materials and to enlighten the new trends in laser technology focused on dental materials management. New approaches for the elaboration of dental materials that require high energy levels and delicate processing, such as metals, ceramics and resins are provided, while time consuming laboratory procedures, such as cutting restorative materials, welding and sintering are facilitated. In addition, surface characteristics of titanium alloys and high strength ceramics can be altered. Finally, the potential of lasers to increase the adhesion of zirconia ceramics to different substrates has been tested for all laser devices, including a new ultrafast generation of lasers influence.
Indeed, the use of lasers in dentistry has recently received much attention, in both clinical practice and research; their unique properties produce favourable clinical results in some cases and encourage patient acceptance. The concept of lasers dates back to 1917 with Einstein’s theory of stimulated emission, but it was not until 1960 that the first working laser was created by Theodore Maiman. Lasers are currently used in a wide range of medical and cosmetic procedures.
However, they have only recently received attention in clinical dental settings. Lasers are being recognized for their ability to ablate hard tissues with minimal anesthesia, reduce bacteria counts in root canals and even provide hemostasis of soft tissues during their use.
The present review describes the current state of laser application in dental materials science in a sufficiently good summary presentation. In the view of the reviewer, there is nothing to add to this presentation. However, the reviewer lacks a focus on the dental material properties in laser application taking into account specific dental clinical pictures, especially when applied under conditions of marginal periodontitis or peri-implantitis. Here, the specific literature research is missing in the review. Various types of lasers have been investigated as an adjunct to periodontal therapy; these include carbon dioxide (CO2), diode, neodymium:yttrium–aluminium–garnet (Nd:YAG) and erbium: yttrium–aluminium–garnet (Er:YAG) lasers. However, adverse results have been associated with each type, including thermal damage to root surfaces, increases in pulpal temperature and the production of toxic by-products. The Er:YAG laser has produced the most promising results, as it can ablate effectively with minimal adverse effects. Obviously, the authors are not aware of the current literature in the field, a quick pubmed search should be sufficiently.
The publication is comprehensive, with detailed pictures and tables on the examination methods and the presentation of results.
Author Response
Thank you for your encouraging comments! Τhis review is focused basically on dental materials and mainly in zirconia, base metal alloys,titanium and resin polymers. For those materials and especially for the new entry –zirconia- only scarce information can be found as far as laser applications are concerned. The revised document is enriched with more details all minor mistakes were corrected and two descriptive Tables were inserted to highlight all available materials and laser assisted procedures respectively.)
Reviewer 3 Report
- The introduction should be improved. It is repetitive, inaccurate and contains unnecessary information and basic mistakes. Excited is not the same than stimulated. The pulse generation is more complex than indicated in the introduction. It is not necessary to define what the word LASER means, nor the dual nature of ligth. E. Skoulas should read carefully the whole manuscript because there are lots of basic errors.
- I do not understand Figure 1. Why power density depends on pulse lenght? It is not possible to correlacionate pulse duration with power density without taking into account the other irradiation parameters. In any cas,e if nanosecond pulses produce material removal by direct ablation, liquid phase formation is not expected for picosecond pulses, as the pulse power density increases.
- sentence in line 85 is not true
- In line 202 it is said that resin removal is due to laser melting of the composite. Is there any evidence of resin melting? A thermostable polymer does not melts under heat, maybe it suffers thermal degradation and detachment from dentine.
- Soldering and welding should not be confused (213)
- Ti6Al4V, please correct in 275
- Sentence 298 - 302 should be rewritten, and the way of expressing the citations, unified with the rest of the manuscript.
- Sentence 304 is confusing, diffusion and melting following of melting again. Please, review it.
- As a general rule giving only the average power is not very indicative (ref 79 and 86, for example). It is very relevant to know other parameters as pulse frequency, spot size, pulse energy, pulse length, etc.
- Sentence 332-334 is not true, the energy released do not depend on beam diameter
- In 347. The laser power is not the energy intensity. I any case what energy intensity is? energy density maybe?
- Figure caption 4, has the reference twice
- Conclusions, correct line 390.
- In future perspectives. The sentence property to deliver... is not clear.
- Several references are not completes (76,87). Reference 41 includes ref 81, 43 includes 90, 70 includes 99,...
Author Response
Reviewer 2
- The introduction should be improved. It is repetitive, inaccurate and contains unnecessary information and basic mistakes. Excited is not the same than stimulated. The pulse generation is more complex than indicated in the introduction. It is not necessary to define what the word LASER means, nor the dual nature of ligth. E. Skoulas should read carefully the whole manuscript because there are lots of basic errors.
We thank the reviewer for the comment the introduction was improved accordingly.
- I do not understand Figure 1. Why power density depends on pulse lenght? It is not possible to correlacionate pulse duration with power density without taking into account the other irradiation parameters. In any cas,e if nanosecond pulses produce material removal by direct ablation, liquid phase formation is not expected for picosecond pulses, as the pulse power density increases.
Fig 1 is not presented to account for all physical processes that take place transiently from the excitation to ablation etc (it is not possible in a single figure), but to give the reader a perspective regarding the temporal characteristics of 3 major pulse durations which are most commonly available commercially. As well as the main physical processes that can take place epigrammaticallyfrom the excitation to morphological changes. To avoid confusion, we amended the text to explain better Fig. 1.
The laser pulse intensity or power density or peak power for a single pulse is strongly dependent on the pulse duration as depending the pulse width it can vary with orders of magnitude (Pulse Intensity = (W/cm2)), please check this reference:Mourou, G. A., Tajima, T., &Bulanov, S. V. (2006). Optics in the relativistic regime. Reviews of Modern Physics, 78(2), 309–371.If the reviewer is referring to fluence or peak fluence (J/cm2) of course it is dependent on the repetition rate the average power and the beam waist.
- sentence in line 85 is not true
It is difficult to understand which part is not true is the reviewer is referring to multi-photon absorption it is one of the most commonly accepted physical models to account for absorption of dielectrics or other high band gap materials with photon energies that are multiples times lower (eV). So given that there are not commercial x-ray or deep UV lasers yet this is the reason we refer to multi-photon absorption.Now if the reviewer means that this can be interestingly defined by Keldysh parameter on whether it is based on tunneling, intermediate or pure multi-photon absorption. We choose not to include these cases as we believe that it is too advanced for the broad and interdisciplinary audience of this work.
- In line 202 it is said that resin removal is due to laser melting of the composite. Is there any evidence of resin melting? A thermostable polymer does not melts under heat, maybe it suffers thermal degradation and detachment from dentine.(it was changed according to reviewers suggestion) We have to state that according to the findings in papers 45 (Nikolinakos et al) posterior composite resin on class I cavities who experienced Er Yag laser ablation, disappeared, probably due to evaporation or total disintegration of the resinous matrix leaving a crater in the area of ablation. Dental composite resins are not extremely thermostable ,because they are designed to suffer only normal temperatures of food and beverages 4-60oC and human body temperatures and not the extremely high temperatures caused by the laser ablation.For brackets debonding there is a possibility to accept the term disintegration leading to detachment from ENAMEL interface.Thermal analysis and structural investigation of different dental composite resins Journal of Thermal Analysis and Calorimetry 2008;94(3):791-796
- Soldering and welding should not be confused (213)(soldering was replaced by connecting)
- Ti6Al4V, please correct in 275(corrected)
- Sentence 298 - 302 should be rewritten, and the way of expressing the citations, unified with the rest of the manuscript.(corrected)
- Sentence 304 is confusing, diffusion and melting following of melting again. Please, review it.(corrected)
- As a general rule giving only the average power is not very indicative (ref 79 and 86, for example). It is very relevant to know other parameters as pulse frequency, spot size, pulse energy, pulse length, etc. (as reviewer suggested other parameters were included )
- Sentence 332-334 is not true, the energy released do not depend on beam diameter(corrected).
- In 347. The laser power is not the energy intensity. I any case what energy intensity is? energy density maybe?(it was corrected energy density)
- Figure caption 4, has the reference twice((corrected)
- Conclusions, correct line 390.(corrected)
- In future perspectives. The sentence property to deliver... is not clear.(corrected)
- Several references are not completes (76,87). Reference 41 includes ref 81, 43 includes 90, 70 includes 99,..(corrected)
We thank the reviewer for the detailed reviewing and we apologize for the confusing mistakes that were due to time pressure and extended literature which is often observed in reviews.
Reviewer 4 Report
Authors introduce a review of LASER applications on dentistry. The aim is to review some applications for medical purpouses. Even if one author has specific Laser background the draft is tailored on medical expertise and does not present a physical- engineering analysis, out of the scope of the draft. The review is supported by a significative bibliography. The introductive historical part is useful being the actual draft a Review as well as the first 4 paragraphs that introduce the principal known applications of laser technology in dentistry.
Paragraph 5 is the more interesting by the scientific viewpoint being applied on the study of Zirconia application, a material that is collecting many attention by the dental community.
The draft is lean and focussed. Paragraphs 6 an 7 give a final prospective on these applications.
Author Response
Thank you for your encouraging comments! Τhis review is focused basically on dental materials and mainly in zirconia, base metal alloys,titanium and resin polymers. For those materials and especially for the new entry –zirconia- only scarce information can be found as far as laser applications are concerned. The revised document is enriched with more details all minor mistakes were corrected and two descriptive Tables were inserted to highlight all available materials and laser assisted procedures respectively.
Round 2
Reviewer 1 Report
The reviewer thinks the manuscript has been revised well. The newly added two tables would need the title and caption though.
Reviewer 3 Report
The paper has been improved with the corrections of the authors and can be published in the present form. It only needs to be reviewed to correct some misprints.